# Predicting Flexible Pavement Distress and IRI Considering Subgrade Resilient Modulus of Fine-Grained Soils Using MEPDG

**DOI:** 10.3390/ma16031126

**Published:** 2023-01-28

**Authors:** Kazi Moinul Islam, Sarah L. Gassman

**Affiliations:** Department of Civil and Environmental Engineering, University of South Carolina, Columbia, SC 29208, USA

**Keywords:** pavement distress, IRI, MEPDG, subgrade resilient modulus, FWD

## Abstract

This paper highlights the subgrade resilient modulus (M_R_), which is recognized as an important parameter to characterize the stiffness of the subgrade soil for designing flexible pavement. In this study, 18 thin-walled Shelby tube samples of fine-grained subgrade soils were collected from two sites in South Carolina (Laurens/SC-72 and Pickens/SC-93) and tested in the laboratory using AASHTO T307-99 to obtain the M_R_. In addition, falling weight deflectometer (FWD) tests were performed on the same pavement sections to obtain the back-calculated M_R(FWD)_ per the AASHTOWare 2017 back-calculation tool. A subgrade M_R_ catalog was established and used to select hierarchical Input Level 2 for Pavement Mechanistic-Empirical design (PMED) analysis (v 2.6.1). The PMED analysis was run for 20 years. The Mechanistic-Empirical Pavement Design Guide (MEPDG) and global calibration values were used to predict asphalt concrete (AC) pavement distresses (e.g., rutting, bottom-up fatigue, top-down fatigue, and transverse cracking) and International Roughness Index (IRI) for each pavement section. The predicted values were compared to the field-measured values to determine bias and the standard error of the estimate to validate each distress prediction model for local calibration.

## 1. Introduction

The Mechanistic-Empirical Pavement Design (MEPDG) is based on mechanistic-empirical principles [1,2,3]. These principles are used to calculate pavement responses (stresses, strains, and deflections), and those responses are used to compute incremental damage over time. The MEPDG and newly updated AASHTOWare Pavement ME Design (PMED) software (v2.6.1) require over 100 inputs to model traffic, climate, materials, and pavement to predict the progression of key pavement distresses and smoothness loss over the pavement design period. Pavement performance/distress is primarily concerned with structural and functional performance. The structural performance relates to its physical condition (such as fatigue cracking and rutting for flexible pavements). Ride quality is the dominant characteristic of functional performance, as measured by the International Roughness Index (IRI). In MEPDG, IRI is estimated incrementally over the entire design period by incorporating distresses such as cracking, rutting, faulting, and punchouts as the major factors influencing the loss of smoothness of pavement [1,2,3].

Numerous studies have been performed to predict pavement distress and IRI for flexible pavement using the MEPDG [4,5,6,7]. Different traffic [8,9,10], materials [11,12,13,14], and climate inputs [5,15,16] have an influence on pavement distress and IRI in the MEPDG. Among the inputs of unbound subgrade materials, the M_R_ significantly affects permanent deformation or pavement rutting [17]. M_R_ represents the pavement’s unbound layer configuration’s stiffness subjected to repeated traffic loading. Typically, higher M_R_ soils show less pavement distress and IRI [6,18]. However, some mixed soils (i.e., silty sands and sandy silts) exhibit high resilience and still yield significant rutting [19]. Therefore, it is necessary to correlate resilient modulus with asphalt concrete (AC) pavement distress and IRI for fine-grained soils.

The study will be focused on flexible pavement performance distress and IRI using three different subgrade resilient modulus. Using currently available PMED software (v2.6.1), the outcome of this study will be compared with the field-measured value to assess the estimate’s local bias and standard error.

## 2. Objectives

The research objectives of this study are to find the potential correlations between asphalt concrete (AC) pavement distresses and subgrade resilient modulus (M_R_) using the AASHTOWare Pavement ME design in South Carolina. The potential correlations can be adapted to create a subgrade resilient modulus catalog for flexible pavement design for South Carolina and sites with similar soil conditions. The pavement distresses to be correlated in this study include permanent deformation of the AC layer, permanent deformation of the total pavement, thermal cracking, bottom-up fatigue cracking, top-down fatigue cracking, and IRI. To fulfill the objectives of this study, two main research questions are addressed in this investigation: 1. How does subgrade M_R_ affect the predicted distresses and IRI for AC design per PMED (v2.6.1)? & 2. How do these predictions change for different asphalt thicknesses?

## 3. Methodology

The main variable to be investigated in this study is the subgrade resilient modulus, M_R_, that will be used to predict pavement distress and IRI using the AASHTOWare PMED software (v2.6.1). The subgrade modulus value was obtained using three different methods: back-calculated M_R_ from field non-destructive test (i.e., FWD) (M_R(FWD)_), laboratory-measured M_R_ conducted in the laboratory (M_R(Lab)_), and using the default value based on soil classification (M_R(Default)_). Non-destructive FWD tests were performed, and M_R_ was back-calculated from the field deflection data [20]. Subgrade soil samples were collected at each FWD test location, and resilient modulus tests were performed using AASHTO T307 [21]. Data for two sites in South Carolina were used in this study (See Figure 1a): Laurens/SC-72 and Pickens/SC-93, where design drawings are available to obtain the pavement layer profile and measurements of pavement distresses have been obtained.

There were 18 coring locations utilized in this study. As shown in Figure 1b,c, 13 coring locations (BH-1 to BH-13) were situated along the 9.8 km pavement section of Laurens/ SC-72, and 5 (BH-1 to BH-5) were located along a 1.9 km section of Pickens/SC-93. At each site, coring began at the “START,” and the pavement was cored at spacings of 305 to 915 m intervals along the pavement surface. Core locations were selected based on the surface appearance of the pavement so that cores could be taken from locations where distresses and no distresses were observed. Subgrade soil samples were collected to a depth ranging from 0.91 to 1.9 m from the bottom of the pavement, depending on site conditions. Boreholes were terminated when the strong gravelly layers were encountered, which made sampling difficult. Thin-walled Shelby tube samples (undisturbed) of the subgrade soil were collected using a jeep jack (see Figure 2a) to remove the Shelby tube at a constant rate in the vertical direction. The tube samples were used to determine the resilient modulus per AASHTO T307. A hand auger collected bulk samples from each borehole (see Figure 2b,c).

The flow chart of the study organization is shown in Figure 3. The traffic, climate, and materials parameters for each site, except for the subgrade M_R_, were obtained and held constant in the analysis. The trial pavement section is analyzed incrementally over time using the pavement response and distress models. The pavement passes if the Achieved Reliability is greater than the Target Reliability. If the reverse is true, the pavement fails. If any key distress/performance criteria fail, the trial design needs to be re-run by increasing the AC layer thickness until the requirements are satisfied. Using the PMED software (v2.6.1), PMED predicts distresses in each trial pavement section based on subgrade MR values and compares them to the measured filed values. Global calibration coefficients were used in this study.

### 3.1. Backcalculated FWD Tests

FWD tests were performed at intervals of approximately 61 m; that is, 157 locations along Laurens/SC-72 and 36 locations along Pickens/SC-93. The FWD tests were performed using the Dynatest system [22]. The apparatus (see Figure 4a,b) consists of seven sensors at seven different offsets (0, 203, 305, 457, 610, 915, and 1194 mm from the loading plate). Each FWD test was performed by applying an impulse load of four different magnitudes (30.5, 40, 54, and 70 kN) and collecting deflection data within the deflection basin [23]. Information on the pavement condition (e.g., layer modulus) can be extracted from the analysis of the deflection data. The layer modulus determined from known FWD data is termed the backcalculated modulus, backcalculated M_R_, or M_R(FWD)_.

### 3.2. Resilient Modulus Tests

Resilient modulus (M_R_) tests were performed in the laboratory on 76 mm diameter by 152 mm long specimens obtained from thin-walled Shelby tubes. A GDS Advanced Dynamic Triaxial Testing System was used to perform the tests per AASHTO T 307. The total load cycle duration for each of the 0-15 sequences (AASHTO T 307) was 1 s, which includes a 0.1-s load duration and a 0.9-s rest period.

For conditioning sequence No. 0, a minimum of 500 repetitions of a load equivalent to a maximum axial stress of 27.6 kPa and corresponding cyclic stress of 24.8 kPa was applied to the specimen (AASHTO T 307). If the sample was still decreasing in height at the end of the conditioning period, stress cycling was continued up to 1000 repetitions. After the initial conditioning was completed, each of the 15 main test sequences was applied in 100 load repetitions. The average recoverable deformation for the last five cycles of each test was recorded. For each of the three confining pressures (41.4 kPa, 27.6 kPa, 13.8 kPa), five different cyclic stresses (12.4 kPa, 24.8 kPa, 37.3 kPa, 49.7 kPa, 62 kPa) were applied to the sample. The generalized constitutive resilient modulus model was used to calculate M_R_ (NCHRP-1-37A, 2004):(1)$$M_{R}=k_{1}P_{a}{\left(\frac{\theta }{P_{a}}\right)}^{k_{2}}{\left(\frac{\tau_{\mathrm{oct}}}{P_{a}}+1\right)}^{k_{3}}$$
where

M_R_ = resilient modulus value

$k_{1}$, $k_{2}$ and $k_{3}$ = model parameters

Pa = atmospheric pressure (101.325 kPa)

θ = bulk stress = (σ_1_ + σ_2_ + σ_3_) = (3σ_3_ + σ_d_)

σ_1_, σ_2_, and σ_3_ = principal stresses and

σ_2_ = σ_3_ and σ_d_ = deviator (cyclic) stress = σ_1_ − σ_3_; and

$\tau_{\mathrm{oct}}$ = octahedral shear stress

= $\sqrt{{\left(\sigma_{1}-\sigma_{2}\right)}^{2}+{\left(\sigma_{1}-\sigma_{3}\right)}^{2}+{\left(\sigma_{2}-\sigma_{3}\right)}^{2}}/3$

After completing the resilient modulus test procedure, the M_R_ was calculated for each sequence per AASHTO T307. For determining $k_{1}$, $k_{2}$, and $k_{3}$, Equation (1) was simplified and transferred to the logarithmic function (See Figure 5). The model parameters were obtained for each test using multiple linear regression techniques and used in Equation (1) to calculate the M_R_ for each specimen.

The laboratory-measured resilient modulus (M_R(Lab)_) was calculated using Equation (1) with confining stress (σ_3_) equal to 13.8 kPa, and cyclic stress (deviator) stress equal to (σ_d_) 41.4 kPa per NCHRP-285 (2004).

### 3.3. Default Resilient Modulus (M_R(Default)_)

The default resilient modulus value was obtained for the unbound materials based on the soil classification [3]. The value of 103 MPa and 90 MPa were used for Laurens/SC-72 and Pickens/SC-93, respectively.

### 3.4. Traffic Inputs

The two-way Average Annual Daily Truck Traffic (AADTT) was calculated using the base year Average Annual Daily Traffic (AADT) and the percentage of trucks annually for the pavement section. There are two lanes in one direction, with an operational speed of 88 km/h and 56 km/h for Laurens/SC-72 and Pickens/SC-72, respectively. The traffic growth rate for each section was assumed to be the same as the historical pavement growth rate (see Table 1). The historical growth rate was determined using AADT measurements from the SCDOT Road Data Services. The vehicle class distribution (FHWA vehicle class 4 through 13), monthly adjustment factors and hourly adjustment factors, axle load configurations (single, tandem, tridem, & quad), and axle per truck values were obtained from the Weigh-in-Motion (WIM) station located at each section. Figure 6 shows an example of axles per truck and vehicle class distribution for the Laurens/SC-72 site. The load values of single axles range from 13 to 182 kN; tandem axles are 27 to 365 kN, and tridem and quad axles are 53 to 454 kN. Default values were used for other traffic inputs (e.g., average axle width, tire pressure, lateral wander, wheelbase, and axle-load configurations) that are not listed in Table 1.

### 3.5. Climate Inputs

The nearest climate stations, 138396 and 138397, were used for Laurens/SC-72 and Pickens/SC-93, respectively. Table 2 shows the climate data for this analysis. Approximately 3 m (10 ft) depth of water table was used as a default value Level 3 for the PMED software.

### 3.6. Materials Inputs

Figure 7 presents the pavement profile for Laurens/SC-93 and Pickens/SC-93. The subgrade input parameters were obtained from laboratory testing (see Table 3) and are classified as Level 1. For unbound subgrade resilient modulus, M_R(Lab)_ and M_R(FWD)_ are Level 2 inputs, whereas M_R(default)_ is a Level 3 input. Each pavement’s average M_R_ value was used to predict distress and IRI. PG 64-22 grade for AC input was selected as a Level 2 input, and dynamic modulus value was used as the default per MEPDG. The remaining AC parameters (see Table 4) were used as Level 3 for each pavement trial.

### 3.7. Performance Criteria/Distress and Reliability

Table 5 [3] shows the performance criteria and reliability levels for the MEPDG. The performance criteria ensure that a pavement design will perform satisfactorily over its project design life. To estimate the initial IRI for each pavement section, linear regression was performed on the data for each pavement section, as shown in Figure 8. The initial IRI was determined by extrapolating the historical IRI plot of each pavement section to the year of pavement construction completion.

## 4. Results and Discussion

Results of the soil classification, gradation, pavement predicted distress, and IRI values are discussed, followed by a thickness sensitivity analysis, compared to field measured and the AASHTOWare PMED predicted values.

### 4.1. Soil Classification and Gradation

Soil classification was conducted according to the USCS (ASTM D 2487) and AASHTO (AASHTO M 145) methods. The #4, #10, #20, #40, #60, #100, and #200 sieves were used to determine the percent finer of the soil from each of the 18 boreholes. Table 6 shows the soil classification and gradation results for 18 samples.

### 4.2. Pavement Distress

Figure 9a shows the AC total permanent deformation for 20 years. For Laurens/SC-72, the rutting at a period of 20 years ranged from 4.9 mm (found from M_R(FWD)_) to 18.2 mm (from M_R(Lab)_), a difference of 115%, and the threshold limit was not met. Similarly, for Pickens/SC-93, the rutting at the age of 20 years ranged from 3.1 mm (found from M_R(FWD)_) to 12.7 mm (from M_R(Lab)_), a difference of 120%, and the threshold limit was not met. This shows that using the laboratory-measured value (M_R(Lab)_) produced the most conservative rutting at 20 years, whereas using the values back-calculated from the FWD data (M_R(FWD)_) and M_R(Default)_ produced the least conservative result. Similar observations are made for a period of 10 years, which is a typical design life for asphalt pavements in South Carolina, given the historical tendency of flexible pavement to deteriorate after approximately 12 to 15 years, regardless of traffic [24].

Figure 9b presents the predicted AC top-down cracking for the two pavement sections. The top-down cracking profile for Laurens/SC-72 increased from approximately 4% to 11% after 20 years of age; whereas, for Pickens/SC-72; the profile with age remained constant at 4.6%, which is well below the threshold of 25%. Similarly, the predicted performance for AC bottom-up cracking shown in Figure 9c followed a similar pattern as AC top-down cracking. The maximum and minimum values were observed at 17% and 1.5% for M_R(Lab)_ and M_R(Default),_ respectively, for the Laurens/SC-72, whereas for Pickens/SC-72, the profile with age remained constant at 1.5. For the range of M_R_ studied herein (M_R(FWD)_, M_R(Lab)_, or M_R(Default)_)_,_ the pavements are adequately designed to prevent rutting, top-down cracking, bottom-up cracking, and transverse cracking.

The value of AC transverse/thermal cracking for the Laurens/SC-72 and Pickens/SC-72 is shown in Figure 9d. Both sections follow a similar pattern for thermal cracking, with age for Laurens/SC-72 and Pickens/SC-93 remaining constant at 41 and 84 m/km, respectively.

It is important to note that while the M_R_ of the subgrade soil was the main parameter investigated in this study, other properties of the asphalt layers can significantly influence rutting. For example, increasing the air voids from 4% to 7% has been shown to decrease stiffness and fatigue life by 25% and 69%, respectively [25], and the resilient modulus of warm-mix-asphalt has been shown to be lower than for hot-mix-asphalt [26]. Additionally, changing Poisson’s ratio and resilient modulus for asphalt aggregate base may have a greater effect on pavement deformation [27]. Future work is necessary to study these parameters in the context of the study herein.

### 4.3. Smoothness/International Roughness Index (IRI)

The predicted value of IRI for the two AC pavement sections is shown in Figure 10. For Laurens/SC-72, the IRI at the age of 20 years ranging from 2662 mm/km (found from M_R(FWD)_ & M_R(Default)_) to 2776 mm/km (from M_R(Lab)_), a difference of 4.2% and the corresponding pavement ages at which the limiting criterion was met were 20 and 18 years, respectively. Similarly, for Pickens/SC-93, the IRI at the age of 20 years ranged from 3025 mm/km (found from M_R(FWD)_ to 3104 mm/km (from M_R(Lab)_), a difference of 2.5% and the corresponding pavement ages at which the limiting criterion was met were 13 and 12 years, respectively. The result shows that using the laboratory-measured value (from M_R(Lab)_) produced the most conservative IRI at 20 years, whereas using the values back-calculated from the FWD data (M_R(FWD)_) or M_R(Default)_ had the least conservative result. For MR(Default), the threshold limit was met at 20 years for Laurens/SC-72 and 13 years for Pickens/SC-93. Based on the predicted IRI, the difference in the age at which the threshold limit was met was not very sensitive to the MR that was used in the analysis (M_R(FWD)_, M_R(Lab)_, or M_R(Default)_). Results also show that Pickens/SC-93 predicted higher IRI compared to Laurens/SC-72 because Pickens/93 initial IRI was 1.4 times higher than Laurens/SC-93. So, Initial IRI is most important to predict terminal IRI using PMED software for the cases studied herein.

Overall, IRI is the controlling factor for designing these two flexible pavements. IRI combines thermal cracking, rutting, and fatigue cracking, among other site-specific factors, and the prediction of IRI in PMED is expressed by the following Equation (2) [1,2,3] as:IRI = IRIo+C1(RD) + C2(FC_Total_) + C3(TC) + C4(SF)(2)
where IRIo = Initial IRI after construction, m/km, SF = Site factor, FC_Total_ = Area of fatigue cracking (combined alligator, longitudinal, and reflection cracking in the wheel path), percent of total lane area. TC = Length of transverse cracking (including the reflection of transverse cracks in existing AC pavements), m/km; RD = Average rut depth, mm; C1,2,3,4 = Global calibration factors: C1 = 40.0, C2 = 0.400, C3 = 0.008, and C4 = 0.015.

As per AASHTO [1,2,3], the laboratory-measured M_R(Lab)_ is considered a hierarchical input Level 2 and is the preferred method to compare predicted results to measured distress values. For M_R(Lab),_ the design threshold value was reached at 18 and 12 years for Laurens/SC-72 and Pickens/SC-93, respectively. Based on the analysis, the section needs to be rehabilitated, or an asphalt thickness layer should be increased to reach the desired pavement service life.

### 4.4. Thickness Sensitivity Analysis Using Different M_R_

Asphalt cores were obtained in conjunction with manual distress surveys at the two sites. A total of 18 cores were obtained: 13 at Laurens/SC-72 and 5 at Pickens/SC-93, and the thickness of each core was measured. Figure 11a shows the core thickness profiles of Laurens/SC-72 and Pickens/SC-93. For Laurens/SC-72, the core thicknesses ranged from 16.5 cm to 36.8 cm, and the values for Pickens/SC-93 ranged from 20.3 cm to 26.7 cm. For sensitivity analysis, the thickness varied from 8 cm to 38 cm at 5 cm intervals, as shown in Figure 11b. The three different M_R_ were used to estimate the IRI for each thickness. For M_R(Lab),_ the minimum thickness was found to be 13 cm for Laurens/SC-72, whereas a minimum thickness was not achieved for the range of MR studied for Pickens/SC-93. The thickness optimization tool in PMED was used to evaluate layer thickness, and thickness was optimized up to 66 cm (26 in). Results show that the predicted IRI exceeded the threshold limit (2715 mm/km or 172 in/mile), and the criteria were not satisfied for this analysis (see Figure 11c). Note that the measured IRI for Pickens/SC-93 was higher than the Laurens/SC-72 (see Table 4). To improve the results and satisfy the criteria, site-specific values of the material properties should be used instead of the default values, and local calibration factors are needed.

### 4.5. Comparison between Measured and Predicted Distress

Table 7 shows a comparison between the average and predicted distresses for Laurens/SC-72 and Pickens/SC-93. The measured values of field distress (AC rutting, top-down cracking, bottom-up cracking, and transverse cracking) were collected in 2018 [28], and the IRI values were obtained from the SCDOT Pavement Management System. Table 4 shows that the average predicted rutting is approximately 5 and 10 times higher than the measured value for Laurens/SC-72 and Pickens/SC-72, respectively. The predicted AC top-down cracking is a close agreement (~5.7% difference) with measured value for Laurens/SC-72, whereas 50% lower for Pickens/SC-93. For AC bottom-up cracking, the predicted value is 21% lower than the measured value for Laurens/SC-72 and 27% higher for Pickens/SC-93. Both pavement sections predicted a lower value than the measured value for transverse cracking. The measured value of IRI exceeded the predicted IRI for all of these subgrade M_R_. Thus, it is concluded that the IRI is over predicted for both sections.

Table 7 also summarizes the residual error/bias (er(mean)) and standard error of estimate (SEE) for both sections. The bias and SEE are low for all distresses except IRI. As stated in Section 4.4, the IRI values heavily depend on the other distresses calculated by the AASHTOWare PMED and site factor. Research is ongoing to eliminate local bias and reduce SEE for developing regional or local calibration factors for predicting pavement distress and IRI, as per AASHTO [29].

## 5. Summary of Findings and Conclusions

This study investigated the flexible pavement distress and IRI considering the three different subgrade resilient modulus (e.g., M_R(Lab),_ M_R(FWD), &_ M_R(Default)_) for two sites with fine-grained soils in South Carolina. The results are summarized below:

AC rutting: Results show that the laboratory-measured M_R(Lab)_ predicted higher rutting than FWD and default values. The maximum rutting was observed at 18.2 mm and 12.5 mm at the end of design life for Laurens/SC-72 and Pickens/SC-93, respectively. The predicted rutting was below the design threshold yet exceeded the field-measured distress. Thus, the AASHTOWare PMED over-predicted the AC rutting for the flexible pavement design.

AC top-down and bottom-up cracking: laboratory-measured M_R(Lab)_ predicted the lowest AC top-down cracking for Laurens/SC-72 compared to M_R(FWD)_ and M_R(Default)_, whereas cracking remained constant with pavement age for all three MR values for Pickens/SC-93. For MR(Lab), the PMED predicted 1.4 times higher bottom-up cracking than the field-measured value for Laurens/SC-72. Pickens/SC-93 predicted value was 1.4%, whereas the field measured value was 1.1%.

Transverse cracking: The predicted transverse cracking was 41% and 84% for Laurens/SC-72 and Pickens/SC-93, respectively. The PMED predicted 75% and 18% lower values than measured transverse cracking for Laurens/SC-72 and Pickens/SC-93, respectively.

Smoothness/IRI: Results show that the higher resilient modulus (M_R(FWD)_) predicted lower IRI and higher pavement service life. The maximum bias and SEE were observed at the IRI prediction, so extensive research is needed to eliminate bias and reduce SEE to determine local calibration factors per AASHTO [29] in South Carolina.

This study also investigated the effect of asphalt thickness on pavement performance. The observations are summarized as follows:

AC rutting: For a range of AC thickness from 8 cm to 38 cm at a 5 cm interval, the PMED analysis for AC rutting showed a reduction of 57% and 62% with increasing asphalt thickness for Laurens/SC-72 and Pickens/SC-93, respectively.

AC bottom-up cracking: For Laurens/SC-72, at an initial thickness of 8 cm, the AC bottom-up cracking was observed around 15%. With increasing AC thickness, the cracking value was significantly reduced and recorded at approximately 1.4% for a thickness of 38 cm. On the contrary, AC bottom-up cracking did not change with increasing AC thickness for Pickens/SC-93. The cracking value was 1.5% for all thicknesses studied.

AC top-down and transverse cracking: For the range of AC thicknesses studied, the predicted top-down cracking and transverse cracking did not change with AC thickness. Top-down cracking was approximately 4% for all thicknesses for both sites, whereas transverse cracking was 41% and 84% for all thicknesses studied for Laurens/SC-72 and Pickens/SC-93, respectively.

Smoothness/IRI: Using M_R(Lab)_, the predicted IRI decreased by 11% and 6% for Laurens/SC-72 and Pickens/SC-93, respectively, for thicknesses from 8 cm to 38 cm.

The findings in this study can help one understand the effects of subgrade resilient modulus variables on distress predictions and IRI and are also beneficial in creating a M_R_ catalog.

## Figures and Tables

**Figure 1 materials-16-01126-f001:**
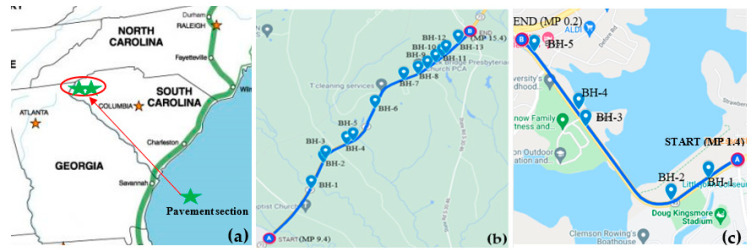
(**a**) Pavement location map and coring Location map for (**b**) Laurens/SC-72 (https://www.google.com/maps/d/u/0/edit?mid=1jj-00vFohfGFblAqnFJDa3s54mlurHyc&ll=34.39052226622161%2C-81.85667629577638&z=13, accessed on 3 April 2022) and (**c**) Pickens/SC-93 (https://www.google.com/maps/d/u/0/edit?mid=16fAsTCLDfDB_4usuSCCZHGEIdSW-Z4Cy&ll=34.680649589059044%2C-82.83778701651015&z=15, accessed on 3 April 2022).

**Figure 2 materials-16-01126-f002:**
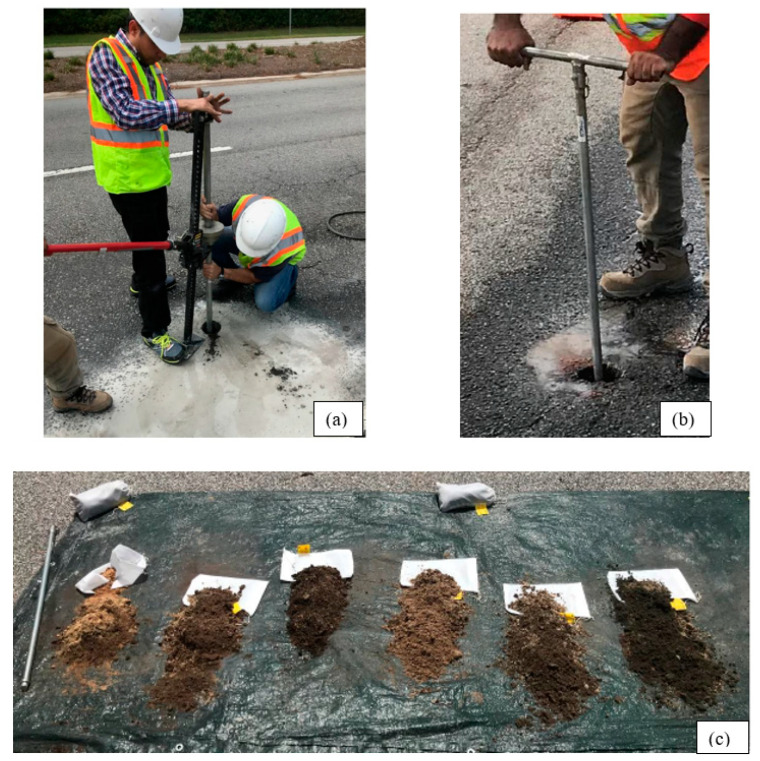
Subgrade sample collection process: (**a**) Shelby tube sample collected with assistance from A Jeep Jack, (**b**) Bulk sample collected by hand auger, and (**c**) Bulk samples.

**Figure 3 materials-16-01126-f003:**
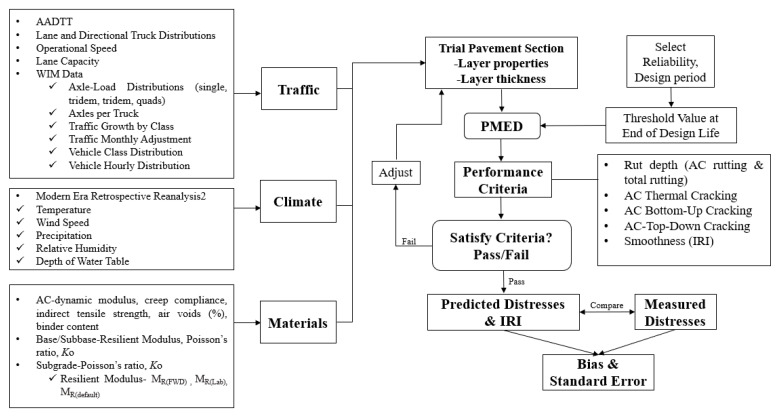
Flow chart of the study.

**Figure 4 materials-16-01126-f004:**
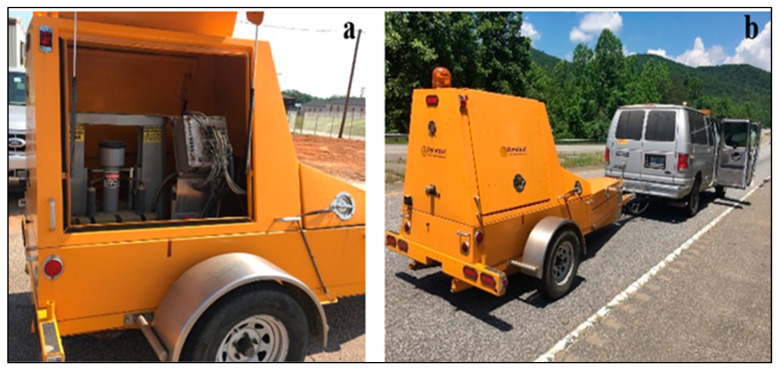
FWD:(**a**) Loading frame, and (**b**) system.

**Figure 5 materials-16-01126-f005:**
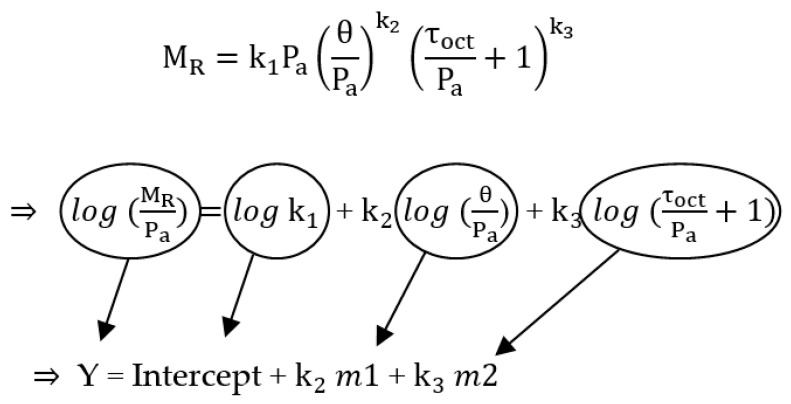
Logarithmic function of Equation (1).

**Figure 6 materials-16-01126-f006:**
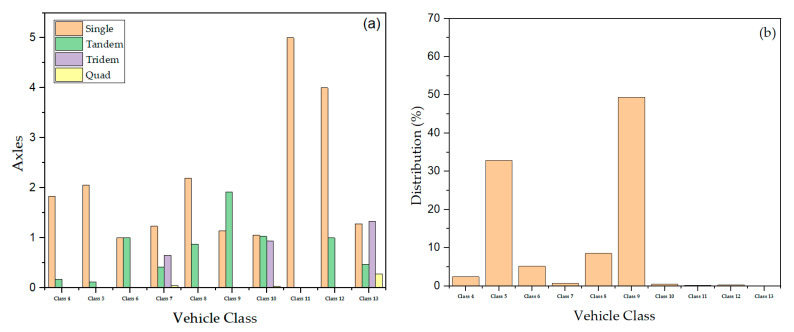
Example of (**a**) axle per truck and (**b**) vehicle class distribution for Laurens/SC-72.

**Figure 7 materials-16-01126-f007:**
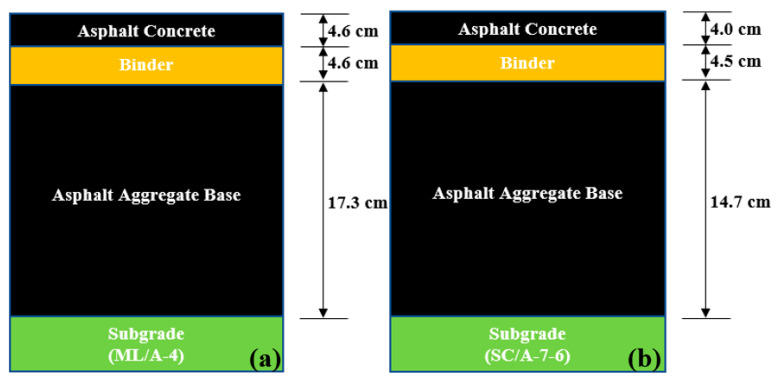
Pavement profile: (**a**) Laurens/SC-72, and (**b**) Pickens/SC-93 (figures are not drawn to scale).

**Figure 8 materials-16-01126-f008:**
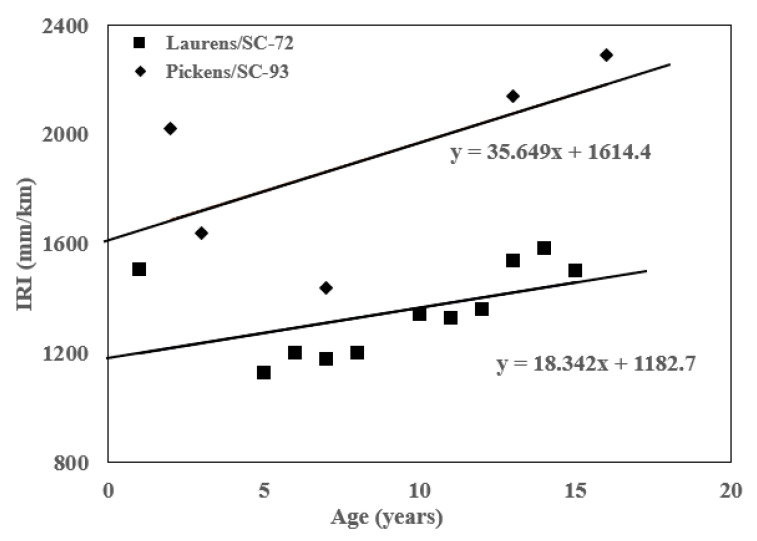
Initial IRI estimation for Laurens/SC-72 and Pickens/SC-93.

**Figure 9 materials-16-01126-f009:**
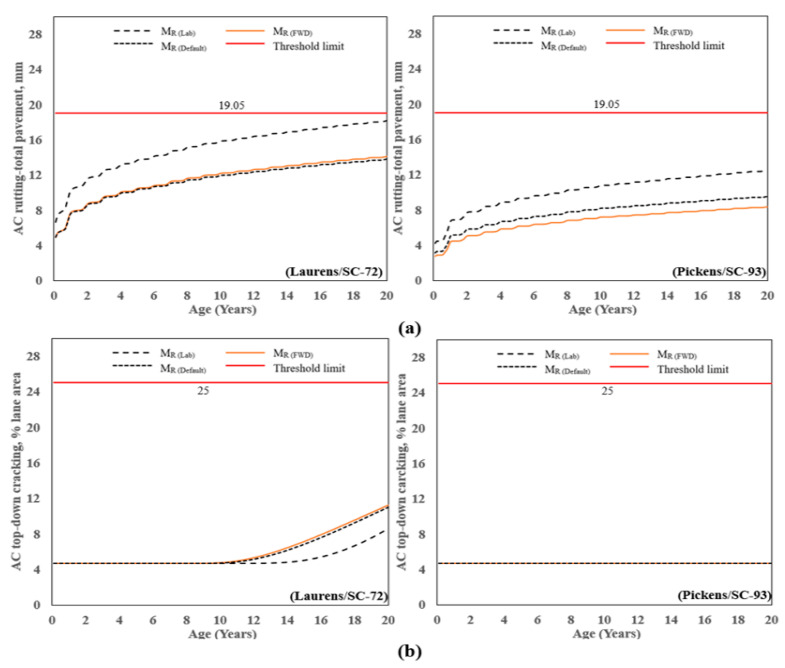
Predicted distress vs. time: (**a**) AC rutting, (**b**) AC top-down cracking, (**c**) AC bottom-up cracking, and (**d**) AC transverse/thermal cracking.

**Figure 10 materials-16-01126-f010:**
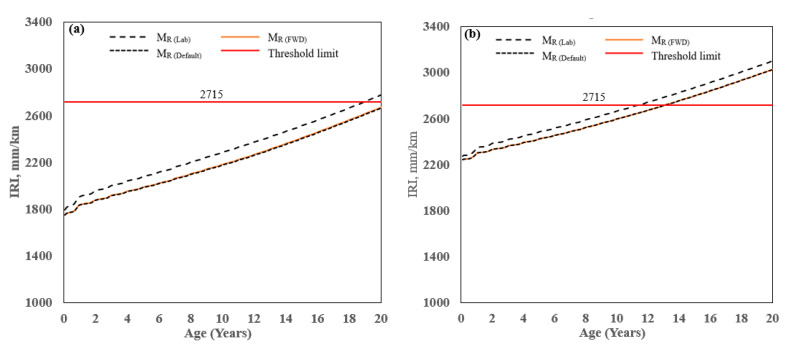
Predicted IRI: (**a**) Laurens/SC-72, and (**b**) Pickens/SC-93.

**Figure 11 materials-16-01126-f011:**
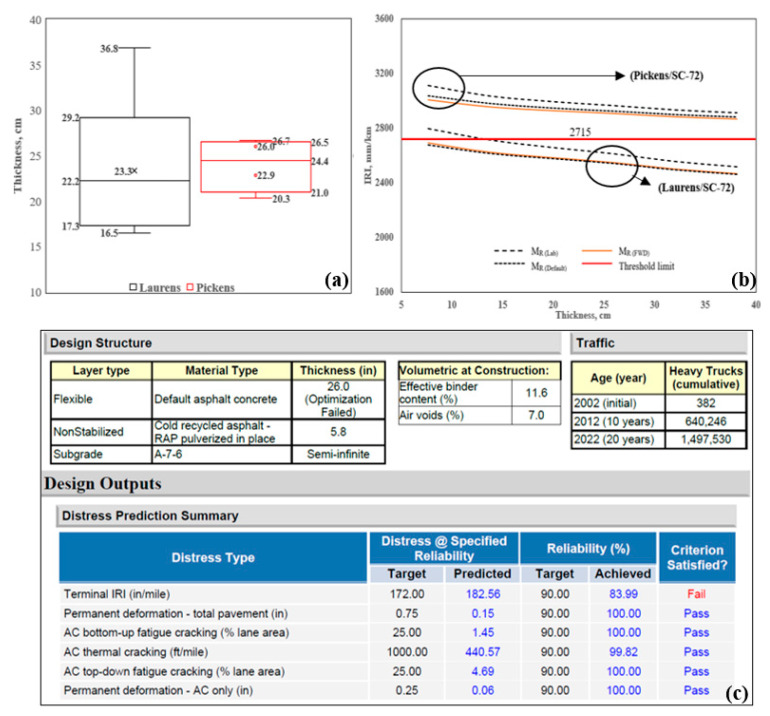
Results for Laurens/SC-72 and Pickens/SC-93: (**a**) Asphalt core thickness profile, (**b**) Sensitivity analysis for predicting IRI, and (**c**) Thickness optimization result for Pickens/SC-93 (picture directly taken from AASHTOWare PMED output).

**Table 1 materials-16-01126-t001:** Traffic inputs for the flexible pavement design.

Input Parameters	Traffic Level	
	Laurens/SC-72	Pickens/SC-93
Initial two-way AADTT	834	382
Number of lanes	2	2
Growth rate (%)	2.04	0.87
Percent truck in design direction	50.3	51.7
Percent truck in design lane	84.1	75.0
Operational speed (km/h)	88	56

**Table 2 materials-16-01126-t002:** Summary of climate data.

Input Parameter	Laurens/SC-72	Pickens/SC-93
Mean annual air temperature (deg F)	55.9	57.8
Mean annual precipitation (in)	65.8	58
Freezing index (deg F days)	104.9	70.8
Average annual number of freeze/thaw cycles	70.4	63
Number of wet days	294.4	289.7

**Table 3 materials-16-01126-t003:** Summary of material inputs for AASHTOWare PMED (v2.6.1).

County/Region			Laurens/SC-72	Pickens/SC-93
Base Year			2002	2001
Design Life (years)			20	
Asphalt Binder **			PG 64-22	
Effective Binder Content (%) ***			11.6	
Unit Weight (kN/m^3^) ***			23.5	
Air Void (%) ***			7	
Dynamic Modulus (MPa) ***			Input level 3	
Subgrade properties				
Avg. Subgrade Modulus (MPa)	Lab	M_R(Lab)_ **	54	45
	FWD	M_R(FWD)_ **	109	137
	Default	M_R(Default)_ ***	103	90
USCS/AASHTO			ML/A-4	SC/A-7-6
Liquid Limit & Plasticity Index *			48 & 20	42 & 14
%F			47.0	44.0
Maximum Dry Unit Weight, γ_dmax_ (kN/m^3^) *			17.0	18.5
Water Content (%) & Specific Gravity *			18 & 2.74	13.8 & 2.71

Note: * = Level 1 input, ** = Level 2 input, *** = Level 3 input, NP = non-plastic, F = %passing #200 sieve, average M_R_ for each site found from repeated load triaxial tests per [21] on tube samples at in situ water content and density [23]. Average back-calculated MR for each site found from FWD tests using AASHTOWare back-calculation tool (v 1.1.2) [20], default value taken from the PMED software (v2.6.1), soil classification as per ASTM D2488 and AASHTO M145.

**Table 4 materials-16-01126-t004:** Summary of material inputs for asphalt aggregate base.

Parameters	Input Values	Remarks
Poison’s ratio	0.35	Default values per MEPDG 2020 [3]
Coefficient of lateral earth pressure	0.5	
Resilient Modulus, MPa	138	
Gradation		
Sieve Size	Percent Passing	
#200	8.7	
#80	12.9	
#40	20	
#10	33.8	
#4	44.7	

**Table 5 materials-16-01126-t005:** Performance criteria and reliability summary for flexible pavement trial design.

Distress Type	Threshold Value at the End of Design Life & Reliability (%)
Initial IRI, mm/km	Laurens/SC-72: 1182.7 & Pickens/SC-93: 1614.4
Terminal IRI, mm/km	2715 & 90
AC top-down fatigue cracking, % lane area	25 & 90
AC bottom-up fatigue cracking, % lane area	25 & 90
AC transverse (thermal) cracking, m/km	189.4 & 90
AC rutting/Permanent deformation–total pavement, mm	19.05 & 90

**Table 6 materials-16-01126-t006:** Soil classification and gradation for 18 samples.

Site	BH	Gradation Size ^1^							USCS/AASHTO ^2^
		#4	#10	#20	#40	#60	#100	#200	
Laurens /SC-72	1	99.2	97.1	91.0	80.2	75.0	51.0	47.0	ML/A-4
	2	95.2	93.8	90.0	72.0	66.5	52.0	52.0	ML/A-4
	3	99.0	97.0	91.0	82.0	73.0	52.0	47.0	ML/A-4
	4	98.1	94.2	90.2	85.1	70.2	62.0	54.0	ML/A-4
	5	97.2	93.5	92.0	75.2	70.1	48.5	55.0	ML/A-4
	6	93.0	88.0	80.0	71.0	62.0	54.0	45.0	ML/A-4
	7	99.0	96.2	91.1	79.5	72.4	50.6	45.0	ML/A-4
	8	99.1	97.3	90.5	75.2	62.1	51.3	52.0	ML/A-4
	9	95.0	94.0	89.0	77.0	67.0	56.0	48.0	ML/A-4
	10	93.0	88.0	80.0	71.0	62.0	54.0	51.0	ML/A-4
	11	97.0	95.0	92.0	84.0	76.0	66.3	48.0	ML/A-4
	12	99.0	97.0	91.0	82.0	73.0	52.0	46.0	ML/A-4
	13	98.0	94.2	90.2	85.1	70.2	62.0	40.0	ML/A-4
Pickens /SC-93	1	93.2	88.0	80.0	71.0	62.0	54.0	44.6	SC/A-7-6
	2	99.3	96.2	91.1	79.5	72.4	50.6	43.8	SM/A-7-6
	3	99.2	97.1	91.0	80.2	75.0	51.0	43.4	SC/A-7-6
	4	93.1	88.0	80.0	71.0	62.0	54.0	51.2	ML/A-4
	5	98.1	96.2	91.1	79.5	72.4	50.6	44.0	SC/A-7-6

^1^ %passing #sieve, ^2^ soil classification as per ASTM D2488 and AASHTO M145.

**Table 7 materials-16-01126-t007:** Summary of measured and predicted distress values.

Distress & IRI	Laurens/SC-72							Pickens/SC-93						
	Distress					er(mean)	SEE	Distress					er(mean)	SEE
	yi	xi						yi	xi					
		1	2	3	Avg.				1	2	3	Avg.		
Rutting (mm)	3.0	18.2	14.1	13.8	15.4	12.3	2.4	1.0	12.5	8.3	8.5	9.8	8.8	2.3
Top-down (% lane area)	10.9	8.5	11.2	11.1	10.3	−0.6	1.8	8.8	4.7	4.7	4.7	4.7	−4.1	0
Bottom-up (% lane area)	11.9	17.1	7.3	5.1	9.8	−2.1	2.4	1.1	1.4	1.4	1.4	1.4	0.3	0.2
Transvers (m/km)	166	41	41	41	41	−126	0	102	84	84	84	84	−19	0
IRI (mm/km)	1515	2604	2541	2525	2557	1042	11	2241	2983	2920	2936	2770	529	9.2

Note: yi = measured value; xi = predicted value; 1, 2, & 3 = predicted value taken from using MR(Lab); MR(FWD); & MR(Default), respectively; er (mean) = residual error/bias = xi − yi; SEE = standard error of estimate = $\sqrt{\frac{\Sigma x_{i}-y_{i}}{n-2}}$, n = no. of observations.

## Data Availability

Not applicable.

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
