# Peer review of "Predicting Flexible Pavement Distress and IRI Considering Subgrade Resilient Modulus of Fine-Grained Soils Using MEPDG"

_materials, 2023, doi:10.3390/ma16031126_

Round 1

Reviewer 1 Report

See attached file.

Reviewer 2 Report

Manuscript ID: materials-2119022

Title:    Predicting Flexible Pavement Distress and IRI considering Subgrade Resilient Modulus of Fine-Grained Soils using MEPDG

Materials

The paper presents an interesting paper related to subgrade resilient modulus of fine-grained soils. The following notes were outlined:

1.        Page 1 – line 42: It is preferred to mention the methods used to measure the flexible pavement distress.

2.        Page 2 – line 50: The following studies may be beneficial. You can also refer to them:

·     Hilal, M.M.; Fattah, M.Y., (2022), “Evaluation of Resilient Modulus and Rutting for Warm Asphalt Mixtures: A Local Study in Iraq”, Applied Science, 12, 12841. https://doi.org/10.3390/app122412841.

·     Al-Dossary, A. A. S., Awed, A. M., Gabr, A. R., Fattah, M. Y., El-Badawy, S. M., (2023), “Performance enhancement of road base material using calcium carbide residue and sulfonic acid dilution as a geopolymer stabilizer”, Construction and Building Materials, Vol. 364, 129959. https://doi.org/10.1016/j.conbuildmat.2022.129959.

·     Fattah, M. Y., Al Helo, K. H. I., Qasim, Z. I., (2016), "Prediction Models for Fatigue Resistance of Local Hot Mix Asphalt", Road Materials and Pavement Design, Vol. 17, 4, pp. 793-809, DOI: 10.1080/14680629.2015.1119711, Taylor and Francis Group.

3.     Figure 2 is not clear.

4.     What are the climate inputs used in the analysis ?

5.     In Table 2, mention the test from which each property was determined.

The specific gravity 2.51 is small.

6.     In Figure 5, choosing age of 20 years for pavement is too long.

Reviewer 3 Report

It has certain scientific value

The research  is of limited value

The innovation of the paper is not too high.

This kind of research is very common.

However, it can be considered for publication after revision.

Round 2

Reviewer 1 Report

The similarity index is 37%, checked by Turnitin sophisticated plagiarism checker application. See the receipt.

Reviewer 3 Report

The author carefully revised and responded all the questionsa and comments.

The quality of the manuscript has been greatly improved.

It is recommended to accept and publish.
